# Improving Dirichlet Prior Network for Out-of-Distribution Example Detection

## Abstract

Determining the source of uncertainties in the predictions of AI systems are important. It allows the users to act in an informative manner to improve the safety of such systems, applied to the real-world sensitive applications. Predictive uncertainties can originate from the uncertainty in model parameters, data uncertainty or due to distributional mismatch between training and test examples. While recently, significant progress has been made to improve the predictive uncertainty estimation of deep learning models, most of these approaches either conflate the distributional uncertainty with model uncertainty or data uncertainty. In contrast, the Dirichlet Prior Network (DPN) can model distributional uncertainty distinctly by parameterizing a prior Dirichlet over the predictive categorical distributions. However, their complex loss function by explicitly incorporating KL divergence between Dirichlet distributions often makes the error surface ill-suited to optimize for challenging datasets with multiple classes. In this paper, we present an improved DPN framework by proposing a novel loss function using the standard cross-entropy loss along with a regularization term to control the sharpness of the output Dirichlet distributions from the network. Our proposed loss function aims to improve the training efficiency of the DPN framework for challenging classification tasks with a large number of classes. In our experiments using synthetic and real datasets, we demonstrate that our DPN models can distinguish the distributional uncertainty from other uncertainty types. Our proposed approach significantly improves DPN frameworks and outperform the existing OOD detectors on CIFAR-10 and CIFAR-100 dataset while also being able to recognize distributional uncertainty distinctly.

## 1 Introduction

Deep neural networks (DNNs) have achieved impeccable success to address various real world tasks (Simonyan & Zisserman, 2014a; Hinton et al., 2012; Litjens et al., 2017). However, despite impressive, and ever-improving performance in various supervised learning tasks, DNNs tend to make over-confident predictions for every input. Predictive uncertainties of DNNs can be confronted from three different factors such as *model uncertainty, data uncertainty* and *distributional uncertainty* (Malinin & Gales, 2018). *Model or epistemic uncertainty* captures the uncertainty in estimating the model parameters, conditioning on training data (Gal, 2016). This uncertainty can be explained away given enough training data. *Data or aleatoric uncertainty* is originated from the inherent complexities of the training data, such as class overlap, label noise, homoscedastic and heteroscedastic noise (Gal, 2016). *Distributional uncertainty or dataset shift* arises due to the distributional mismatch between the training and test examples (Quionero-Candela et al., 2009; Malinin & Gales, 2018). In this case, as the network receives unfamiliar *out-of-distribution (OOD)* test data, it should not confidently make predictions. The ability to separately model these three types of predictive uncertainty is important, as it enables the users to take appropriate actions depending on the source of uncertainty. For example, in the active learning scenario, distributional uncertainty indicates that the classifier requires additional data for training. On the other hand, for various real-world applications where the cost of an error is high, such as in autonomous vehicle control, medical, financial and legal fields, the source of uncertainty informs whether an input requires manual intervention.

Recently notable progress has been made to detect OOD images. Bayesian neural network based approaches conflate the distributional uncertainty through model uncertainty (Hernandez-Lobato &

Adams, 2015; Gal, 2016). However, since obtaining the true posterior distribution for the model parameters are intractable, the success of these approaches depends on the chosen prior distribution over parameters and the nature of approximations. Here, the predictive uncertainties can be measured by using an an ensemble of multiple stochastic forward passes using dropouts from a single DNN (Monti-Carlo Dropout or MCDP) (Gal & Ghahramani, 2016) or by ensembling results from multiple DNNs (Lakshminarayanan et al., 2017) and computing their mean and spread. On the other hand, most of the non-Bayesian approaches model their distributional uncertainty with data uncertainty. These approaches can explicitly train the network in a multi-task fashion incorporating both in-domain and OOD examples to produce sharp and flat predictive posteriors respectively (Lee et al., 2018a; Hendrycks et al., 2019). However, none of these approaches can robustly determine the source of uncertainty. Malinin & Gales (2018) introduced Dirichlet Prior Network (DPN) to distinctly model the distributional uncertainty from the other uncertainty types. A DPN classifier aims to produce sharp distributions to indicate low-order uncertainty for the in-domain examples and flat distributions for the OOD examples. However, their complex loss function, using the Kullback-Leibler (KL) divergence between Dirichlet distributions, results in the error surface to become poorly suited for optimization and makes it difficult efficiently train the DNN classifiers for challenging datasets with a large number of classes (Malinin & Gales, 2019).

In this work, we aim to improve the training efficiency of the DPN framework by proposing a novel loss function that also allows the distributional uncertainty to be modeled distinctly from both data uncertainty and model uncertainty. Instead of explicitly using Dirichlet distributions in the loss function, we propose to apply the standard cross-entropy loss on the softmax outputs along with a novel regularization term for the logit (pre-softmax activation) outputs. We train the models in a multi-task function by leveraging both in-domain training images and OOD training images. The proposed loss function can be also viewed from the perspective of the non-Bayesian frameworks (Lee et al., 2018a; Hendrycks et al., 2019) where the proposed regularizer presents an additional term to control the sharpness of the output Dirichlet distributions. In our experiments, we demonstrate that our proposed regularization term can effectively control the sharpness of the output Dirichlet distributions from the DPN to detect distributional uncertainties along with making the framework scalable for more challenging datasets. We also find that the OOD detection performance of our DPN models often improves by augmenting Gaussian noises with the OOD training images to train the DPN. Our experimental results on CIFAR-10 and CIFAR-100 suggest that our proposed approach significantly improves the performance of the DPN framework for OOD detection and out-performs the recently proposed OOD detection techniques.

## 2 RELATED WORKS

In Bayesian frameworks, the predictive uncertainty of a classification model, trained on a finite dataset, $\mathcal{D}_{in} = \{\boldsymbol{x}_i, y_i\}_{i=1}^N \sim P_{in}(\boldsymbol{x}, y)$, is expressed in terms of data (aleatoric) and model (epistemic) uncertainty (Gal, 2016). For an input $\boldsymbol{x}^*$, the predictive uncertainty is expressed as:

$$p(\omega_c|\boldsymbol{x}^*, \mathcal{D}_{in}) = \int p(\omega_c|\boldsymbol{x}^*, \theta) \; p(\boldsymbol{\theta}|\mathcal{D}_{in}) \; d\boldsymbol{\theta} \qquad (1)$$

Here, $\boldsymbol{x}$ and $y$ represents the images and the corresponding class labels, sampled from an underlying probability distribution $p_{in}(\boldsymbol{x}, y)$. Here, the data uncertainty, $p(\omega_c|\boldsymbol{x}^*, \theta)$ is described by the posterior distribution over class labels given model parameters, $\theta$ and model uncertainty, $p(\boldsymbol{\theta}|\mathcal{D}_{in})$, is given by the posterior distribution over parameters given the data, $\mathcal{D}_{in}$.

The expected distribution for predictive uncertainty, $p(\omega_c|\boldsymbol{x}^*, \mathcal{D}_{in})$ is obtained by marginalizing out $\boldsymbol{\theta}$. However, true posterior for $p(\boldsymbol{\theta}|\mathcal{D}_{in})$ is intractable. Approaches such as Monte-Carlo dropout (MCDP) (Gal & Ghahramani, 2016), Langevin Dynamics (Welling & Teh, 2011), explicit ensembling (Lakshminarayanan et al., 2017) approximate the integral in eq. 1 as:

$$p(\omega_c|\boldsymbol{x}^*, \mathcal{D}_{in}) \approx \frac{1}{M} \sum_{m=1}^{M} p(\omega_c|\boldsymbol{x}^*, \boldsymbol{\theta}^{(m)}) \qquad \boldsymbol{\theta}^{(m)} \sim q(\boldsymbol{\theta}) \qquad (2)$$

where, $\boldsymbol{\theta}^{(m)}$ is sampled from an explicit or implicit variational approximation, $q(\boldsymbol{\theta})$ of the true posterior $p(\boldsymbol{\theta}|\mathcal{D}_{in})$. Each $p(\omega_c|\boldsymbol{x}^*, \theta^{(m)})$ represents a categorical distribution, $\boldsymbol{\mu} = [\mu_1, \cdots, \mu_k] = [p(y = \omega_1), \cdots, p(y = \omega_K)]$ over the class labels and the ensemble can be visualized as a collection of points on the simplex. While for a confident prediction, the ensemble is expected to sharply

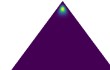 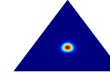 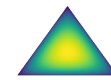 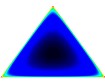

(a) Confident prediction     (b) Data uncertainty     (c) Distributional uncertainty

Figure 1: Desired behavior of a DPN to indicate the three different uncertainties.

appear in one corner of the simplex, the flatly spread ensembles cannot determine whether the uncertainty is due to data or distributional uncertainty. Furthermore, for standard DNNs, with millions of parameters, it becomes even harder to find an appropriate prior distribution and inference scheme to estimate the posterior distribution of the model. Dirichlet Prior Network (DPN) is introduced to explicitly model the distributional uncertainty by parameterizing a Dirichlet distribution over a simplex (Malinin & Gales, 2018). More discussions about DPN is presented in section 3.1.

Alternatively, non-Bayesian frameworks derive their measure of uncertainties using the predictive posteriors obtained from DNNs. Lee et al. (2018a) and Hendrycks et al. (2019) introduce new components in their loss functions to explicitly incorporate OOD data for training. DeVries & Taylor (2018) append an auxiliary branch onto a pre-trained classifier to derive the OOD score. Shalev et al. (2018) uses multiple semantic dense representations as the target label to train the OOD detection network. Several recent works such as (Lee et al., 2018b; Liang et al., 2018) have demonstrated that by tweaking the input images during inference using adversarial perturbations can enhance the performance of a DNN for OOD detection (Goodfellow et al., 2014b). However, their discriminative scores are achieved by tailoring the parameters for each OOD distributions during test time, which is not possible for real-world OOD examples. Hein et al. (2019) propose an adversarial training (Madry et al., 2018) like approach to produce lower confident predictions for OOD examples. However, while these models can identify the total predictive uncertainties, they can not robustly determine whether the source of uncertainty is due to an in-domain input in a region of class overlap or an OOD example far away from the training distribution.

## 3 PROPOSED METHODOLOGY

This section first describes the DPN framework and the difficulties of the existing modeling techniques to scale DPNs for challenging datasets. We then present our improved version DPN by proposing a novel loss function to address these difficulties while allowing to model the distributional uncertainty distinctly from the model and data uncertainty.

### 3.1 DIRICHLET PRIOR NETWORK

A DPN for classification directly parametrizes a prior Dirichlet distribution over the categorical output distributions on a simplex (Malinin & Gales, 2018). For in-domain examples, a DPN attempts to produce sharp Dirichlet in one corner of the simplex, when it is confident in its predictions (Figure 1a). It should produce a sharp distribution in the middle of the simplex to indicate the data (low-order) uncertainty for the in-domain example with a high degree of noise or belongs to a class overlapping region (Figure 1b). In contrast, for OOD examples, a DPN should produce a Dirichlet that spreads over the simplex or across the edge of the simplex to indicate the *distributional uncertainty* (Figure 1c). Here, the *data uncertainty* is expressed by the point-estimate categorical distribution $\boldsymbol{\mu}$ while the distributional uncertainty is described using the distribution over the *predictive categorical* i.e $p(\boldsymbol{\mu}|\boldsymbol{x}^*, \theta)$. The overall predictive uncertainty is expressed as:

$$p(\omega_c|\boldsymbol{x}^*, \mathcal{D}) = \int \int p(\omega_c|\boldsymbol{\mu}) \ p(\boldsymbol{\mu}|\boldsymbol{x}^*, \boldsymbol{\theta}) \ p(\boldsymbol{\theta}|\mathcal{D}) \ d\boldsymbol{\mu} \ d\boldsymbol{\theta} \tag{3}$$

This expression forms a three layered hierarchy of uncertainties: a large model uncertainty, $p(\boldsymbol{\theta}|\mathcal{D})$ would induce a large variation in distributional uncertainty in $p(\boldsymbol{\mu}|\boldsymbol{x}^*, \boldsymbol{\theta})$ and a large degree of uncertainty for $\boldsymbol{\mu}$ leads to higher data uncertainty. DPN framework is consistent with the existing approaches, where an additional layer of uncertainty is included to capture the distributional uncertainty. For example, marginalization of $\boldsymbol{\mu}$ in Eqn. 3 will reproduce Eqn. 1 while loose the control over the sharpness of the output Dirichlet distributions. The marginalization of $\boldsymbol{\theta}$ produces the expected estimation of data and distributional uncertainty given model uncertainty as:

$$p(\omega_c|\boldsymbol{x}^*, \mathcal{D}) = \int p(\omega_c|\boldsymbol{\mu}) \left[ \int p(\boldsymbol{\mu}|\boldsymbol{x}^*, \boldsymbol{\theta}) \ p(\boldsymbol{\theta}|\mathcal{D}) d\boldsymbol{\theta} \right] d\boldsymbol{\mu} = \int p(\omega_c|\boldsymbol{\mu}) \ p(\boldsymbol{\mu}|\boldsymbol{x}^*, \mathcal{D}) d\boldsymbol{\mu} \tag{4}$$

However, similar to eq. 1 marginalizing $\boldsymbol{\theta}$ is eq. 4 is also intractable. Since the model uncertainty is reducible given large training data, for simplicity, here we assume a dirac delta estimation, $\hat{\boldsymbol{\theta}}$ for $\boldsymbol{\theta}$:

$$p(\boldsymbol{\theta}|\mathcal{D}) = \delta(\boldsymbol{\theta} - \hat{\boldsymbol{\theta}}) \implies p(\boldsymbol{\mu}|\boldsymbol{x}^*, \mathcal{D}) \approx p(\boldsymbol{\mu}|\boldsymbol{x}^*, \hat{\boldsymbol{\theta}}) \tag{5}$$

**Constructing a DPN.** A DPN constructs a Dirichlet distribution as a prior over the categorical distributions, which is parameterized by the concentration parameters, $\boldsymbol{\alpha} = \alpha_1, \cdots, \alpha_K$.

$$Dir(\boldsymbol{\mu}|\boldsymbol{\alpha}) = \frac{\Gamma(\alpha_0)}{\prod_{c=1}^{K} \Gamma(\alpha_c)} \prod_{c=1}^{K} \mu_c^{\alpha_c - 1}, \qquad \alpha_c > 0, \quad \alpha_0 = \sum_{c=0}^{K} \alpha_c \tag{6}$$

where, $\alpha_0$ is called the *precision* of the Dirichlet. A larger value of $\alpha_0$ produces sharper distributions to indicate low order uncertainties (fig 1a and 1b). A DPN, $f_{\hat{\theta}}$ produces the concentration parameters, $\boldsymbol{\alpha}$ and the posterior over class labels, $p(\omega_c|\boldsymbol{x}^*; \hat{\boldsymbol{\theta}})$, is given by the mean of the Dirichlet.

$$\boldsymbol{\alpha} = f_{\hat{\theta}}(\boldsymbol{x}^*) \qquad p(\boldsymbol{\mu}|x^*; \hat{\theta}) = Dir(\boldsymbol{\mu}|\boldsymbol{\alpha}) \qquad p(\omega_c|\boldsymbol{x}^*; \hat{\boldsymbol{\theta}}) = \int p(\omega_c|\boldsymbol{\mu}) \ p(\boldsymbol{\mu}|x^*; \hat{\boldsymbol{\theta}}) \ d\boldsymbol{\mu} \ = \frac{\alpha_c}{\alpha_0} \tag{7}$$

A standard DNN with the softmax activation function can be represented as a DPN where the concentration parameters are $\alpha_c = e^{z_c(\boldsymbol{x}^*)}$; $z_c(\boldsymbol{x}^*)$ is the pre-softmax (logit) output corresponding to the class, $c$ for an input $\boldsymbol{x}^*$. The expected posterior probability of class label $\omega_c$ is given as:

$$p(\omega_c|\boldsymbol{x}^*; \hat{\boldsymbol{\theta}}) \ = \frac{\alpha_c}{\alpha_0} = \frac{e^{z_c(\boldsymbol{x}^*)}}{\sum_{c=1}^{K} e^{z_c(\boldsymbol{x}^*)}} \tag{8}$$

However, the mean of the Dirichlet is now *insensitive* to any arbitrary scaling of $\alpha_c$. Hence, the precision of the Dirichlet, $\alpha_0$, degrades under the standard cross-entropy loss.

Malinin & Gales (2018) instead introduced a new loss function that explicitly minimizes the KL divergence between the output Dirichlet and a target Dirichlet to produce a predefined target precision value for the output Dirichlet distributions. For *in-domain examples*, the target distribution is chosen to be a *sharp* Dirichlet, $Dir(\boldsymbol{\mu}|\hat{\boldsymbol{\alpha}}_y)$, focusing on their ground truth classes. While for *OOD examples*, a *flat* Dirichlet, $Dir(\boldsymbol{\mu}|\tilde{\boldsymbol{\alpha}})$ is selected that spreads over the whole simplex.

$$\mathcal{L}(\theta) = \mathbb{E}_{P_{in}} KL[Dir(\boldsymbol{\mu}|\hat{\boldsymbol{\alpha}}_y)||p(\boldsymbol{\mu}|\boldsymbol{x}, \boldsymbol{\theta})] + \mathbb{E}_{P_{out}} KL[Dir(\boldsymbol{\mu}|\tilde{\boldsymbol{\alpha}})||p(\boldsymbol{\mu}|\boldsymbol{x}, \boldsymbol{\theta})] \tag{9}$$

where, $P_{in}$ and $P_{out}$ are the underlying distribution of in-domain and OOD training examples respectively. However, learning the model using sparse 1-hot continuous distributions for class labels, which are effectively a delta function, is challenging due to their complex loss function (eq. 9). Here, the error surface becomes poorly suited for optimization using the back-propagation algorithm (Malinin & Gales, 2019). Malinin & Gales (2018) tackle this problem by using label smoothing (Szegedy et al., 2016) or teacher-student training (Hinton et al., 2015a) to redistribute a small amount of probability density to each corner of the Dirichlet. This technique is found to work well for datasets with a fewer number of class labels. However, for more challenging datasets with a large number of classes, even these techniques cannot efficiently redistribute the probability densities at each corner and results in the target distribution to tend to a delta function. Hence, it becomes difficult to train the DPN to achieve competitive performances. Malinin & Gales (2019) have recently proposed to reverse the terms within the KL divergence in eq. 9 to improve the training efficiency of DPN models. This approach still requires to explicitly constrain the precision of the output Dirichlet distributions using an appropriately chosen hyper-parameter for training.

### 3.2   IMPROVED DIRICHLET PRIOR NETWORK

We now propose an improved technique to model DPN by proposing a novel loss function using the standard cross-entropy loss along with a regularization term to control the precision of the output Dirichlet distribution from the network. As we have seen in equation 8, the precision of a Dirichlet distribution produced by the standard DNN is given as $\sum_{c=1}^{K} \exp z_c(\boldsymbol{x}^*)$. Hence, we can control the sharpness of the distribution by designing a regularization term that increases the sum of logit (presoftmax) outputs for the in-domain examples to produce sharp distributions to indicate their lower uncertainties. For the OOD examples, the regularization term aims to decrease the sum of logit (presoftmax) outputs to produce flat distributions to indicate distributional (higher-order) uncertainties. Hence, instead of explicitly constraining the precision the output Dirichlet with a specific hyperparameter, we allow the network to appropriately produce the precision values for different input.

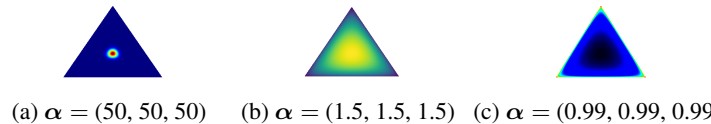

(a) $\boldsymbol{\alpha} = (50, 50, 50)$    (b) $\boldsymbol{\alpha} = (1.5, 1.5, 1.5)$    (c) $\boldsymbol{\alpha} = (0.99, 0.99, 0.99)$

Figure 2: Dirichlet distributions become flatter as the precision is reduced (from (a) to (b)) and the densities move to the edges as the concentration parameters become fractional i.e $\alpha_c \in (0, 1)$.

In this paper, we propose the regularization term as $\frac{1}{K} \sum_{c=1}^{K} \mathrm{sigmoid}(z_c(\boldsymbol{x}))$ to control the sharpness of the output Dirichlet distribution along with the standard cross-entropy loss for classification. Here, the sigmoid function is chosen for the regularization term as it always produces a value within the range of $(0, 1)$ for any $z_c(\boldsymbol{x})$. For in-domain training examples, the loss function is given as:

$$\mathcal{L}_{in}(\boldsymbol{\theta}) = \mathbb{E}_{P_{in}}\left[ -\log p(y|\boldsymbol{x}, \boldsymbol{\theta}) - \frac{\lambda_{in}}{K} \sum_{c=1}^{K} \mathrm{sigmoid}(z_c(\boldsymbol{x})) \right], \tag{10}$$

For OOD training examples, the loss function is given as:

$$\mathcal{L}_{out}(\boldsymbol{\theta}) = \mathbb{E}_{P_{out}}\left[ \mathcal{H}_c(\mathcal{U}; p(\boldsymbol{\omega}|\boldsymbol{x}, \boldsymbol{\theta})) - \frac{\lambda_{out}}{K} \sum_{c=1}^{K} \mathrm{sigmoid}(z_c(\boldsymbol{x})) \right], \tag{11}$$

where, $\mathcal{U}$ denotes the uniform distribution over the class labels. $\mathcal{H}_c$ is the cross-entropy function. We train the network in a multi-task fashion with the overall loss function as:

$$\min_{\boldsymbol{\theta}} \mathcal{L}(\boldsymbol{\theta}) = \mathcal{L}_{in}(\boldsymbol{\theta}) + \lambda \mathcal{L}_{out}(\boldsymbol{\theta}), \qquad \lambda > 0 \tag{12}$$

In this loss function, we have incorporated three user-defined hyper-parameters: $\lambda_{in}$, $\lambda_{out}$ and $\lambda$ in Eq. 10, Eq. 11, and Eq. 12 respectively. $\lambda$ balances between the loss values for in-domain examples and OOD examples. The hyper-parameters $\lambda_{in}$ and $\lambda_{out}$ controls the sharpness of the output Dirichlet from a DPN. By choosing $\lambda_{in} > 0$, we enforce the network to produce positive logit values for in-domain examples that lead to producing sharper Dirichlet distributions (Fig. 2(a)). We choose $\lambda_{in} > \lambda_{out}$ to ensure that the density is either spread over the the simplex or across the boundary (Fig 1c). The choice of $\lambda_{in} > \lambda_{out} > 0$ will lead the network to produce comparatively flatter Dirichlet distributions for OOD examples (Fig. 2b). In contrast, by choosing $\lambda_{out} < 0$, we enforce the network to produce negative values for $z_c(\boldsymbol{x}^*)$ and hence fractional values for $\alpha_c$'s (i.e $\alpha_c \in (0, 1)$) for OOD examples. This will cause the densities of the Dirichlet to be distributed in the edges of the simplex and produces an extremely sharp distribution, as shown in Fig 2c.

The proposed loss function in Eq. 12 is also very closely related to non-Bayesian approaches, where by choosing $\lambda_{in}, \lambda_{out}$ to zero we re-obtain similar loss functions as proposed by Lee et al. (2018a); Hendrycks et al. (2019). However, by setting $\lambda_{in}, \lambda_{out}$ to zero, we lose control over the precision of the Dirichlet distribution that distinguishes distributional uncertainty from data uncertainty.

Our multi-task loss function (eq. 12) requires training samples from the in-domain distribution, $P_{in}$ as well as from OOD $P_{out}$. However, since $P_{out}$ is unknown, Lee et al. (2018a) propose to synthetically generate the OOD training samples from the boundary of in-domain region, $P_{in}$ using generative models such as GAN (Goodfellow et al., 2014a). Alternatively, a different, easily available, real datasets can be used as OOD training examples. In practice, the latter approach is to found to be more effective for training the OOD detectors and has been applied for our experiments on vision datasets (Hendrycks et al., 2019).

## 4 EXPERIMENTAL STUDY

We demonstrate the effectiveness of the DPN framework using the proposed loss function by conducting two sets of experiments. First, we experiment on a synthetic dataset. Next, we present a comparative study of our proposed framework with the existing approaches on CIFAR10 and CIFAR100, and show the advantages over the original DPN framework (Malinin & Gales, 2018).

### 4.1 SYNTHETIC DATASET

We construct a simple dataset with three classes where the instances are sampled from three different isotropic Gaussian distributions as shown in Figure 3(a). We select isotropic co-variances, $\sigma^2 I$ with $\sigma = 4$, to ensure that the classes are overlapping. We train a small DPN with 2 hidden layers of 50

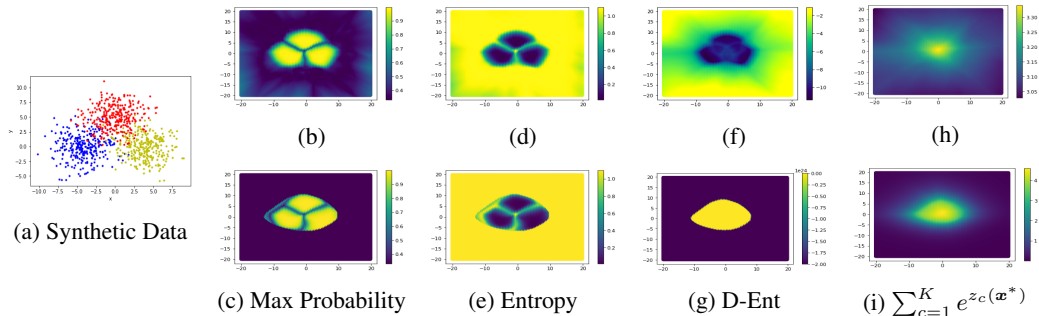

Figure 3: Visualizing the uncertainty measures for different data-points for DPN($\beta = 0.0$) in the top row and for DPN($\beta = 0.5$) in the bottom row.

nodes each for the synthetic dataset. For our loss function, we set $\lambda$ as 1.0. We choose both positive and negative values for $\lambda_{out}$ in our experiments. Here, $\lambda_{in}$ and $\lambda_{out}$ are chosen as $\lambda_{in} = (1 - \beta)$ and $\lambda_{out} = (\frac{1}{\#class} - \beta)$. We train two different sets of DPN models using $\beta = 0.0$ (i.e positive $\lambda_{out}$) and 0.5 (i.e negative $\lambda_{out}$), denoted as DPN($\beta = 0.0$) and DPN($\beta = 0.5$), respectively (See Appendix B for additional details).

An uncertainty measure can be computed as the probability of the predicted class or *max probability*, $max\mathcal{P} = \max_c p(\omega_c|\boldsymbol{x}^*, D_{in})$, in the expected predictive categorical distribution, $p(\omega_c|\boldsymbol{x}^*, D_{in})$ (Figure 3(b) and 3(c) for DPN($\beta = 0.0$) and DPN($\beta = 0.5$), respectively).

*Entropy* of the predicted distribution, $\mathcal{H}[p(\omega_c|\boldsymbol{x}^*, D_{in})] = -\sum_{c=1}^{K} p(\omega_c|\boldsymbol{x}^*, D_{in}) \ln p(\omega_c|\boldsymbol{x}^*, D_{in})$, can be also applied as a total uncertainty measure that produces low scores when the model is confident in its prediction (Figure 3(d) and 3(e) for DPN($\beta = 0.0$) and DPN($\beta = 0.5$) respectively).

*Max probability* and *entropy* are the most frequently used uncertainty measures used by the existing OOD detection models. However, since the predicted distribution is obtained by marginalizing $\boldsymbol{\mu}$ (eq. 3), these measures cannot capture the sharpness of the output Dirichlet for a DPN. Hence, they cannot distinguish between misclassified examples and OOD examples. This observation also indicates the limitation of the existing non-DPN approachesto differentiate between data and distributional uncertainties (Malinin & Gales, 2018). A DPN framework address this limitation by using the *differential entropy* (D-Ent) as an uncertainty measure that produces lower scores for sharper Dirichlet distributions (eq. 13). Appendix D presents the expression to compute D-Ent scores for a given data point.

$$\mathcal{H}[p(\boldsymbol{\mu}|\boldsymbol{x}^*, D_{in})] = -\int_{S^{K-1}} p(\boldsymbol{\mu}|\boldsymbol{x}^*, D_{in}) \ln p(\boldsymbol{\mu}|\boldsymbol{x}^*, D_{in}) \qquad (13)$$

Figure 3(f) and 3(g) demonstrate that D-Ent can distinguish between in-domain and OOD examples. DPN($\beta = 0.0$) produces smaller positive logit values, resulting in flat Dirichlet distributions for OOD examples. Hence, we obtain relatively lower D-Ent scores for OOD examples compared to the in-domain examples (fig 3 (d)). Subsequently, DPN($\beta = 0.5$) produces negative logit values, resulting in sharp Dirichlet distributions across the edge of the simplex for OOD examples (Fig. 2(c)). As we can see in Figure 3(g), the D-Ent scores for OOD examples are even smaller than the in-domain confident predictions, indicating that the output Dirichlet distributions for OOD examples are even sharper.

Since we explicitly constrain the logit outputs to produce smaller values for OOD examples, it is meaningful to define the *sum of the exponential of logits*, $\sum_{c=1}^{K} e^{z_c(\boldsymbol{x}^*)}$, as a new measure of predictive uncertainty. In Figure 3(d) and 3(h), we visualize this uncertainty measure for our DPN($\beta = 0.0$) and DPN($\beta = 0.5$), respectively. We found that our DPN models produce very high logit values. Hence, we scaled down these values by a factor of 100 before computing this uncertainty measure for better visualization. We can see, $\sum_{c=1}^{K} e^{z_c(\boldsymbol{x}^*)}$ produces high scores for all in-domain data points to distinguish them from OOD examples.

## 4.2 EXPERIMENTS ON CIFAR-10 AND CIFAR-100

**Experimental setup.** In this section, we present our experiments on CIFAR-10 and CIFAR-100 datasets (Krizhevsky, 2009). CIFAR-10 images belong to 10 different classes while CIFAR-100 is a more challenging dataset, containing 100 image classes. Both of these datasets contain $32 \times 32$ natural colored images of $50,000$ training and $10,000$ testing examples. In Appendix-A, we have also experimented on TinyImageNet (TIM) that consists of $64 \times 64$ natural images from 200 classes.

The CIFAR-10 classifiers are trained using CIFAR-10 training images as in-domain and CIFAR-100 training images as OOD dataset (Simonyan & Zisserman, 2014b). We use VGG-16 architecture for this case. For

Table 1: Training details for our proposed DPN models

| Classification Task | Input Shape | #Classes | Details of Training data | | Details of Test data |
|---|---|---|---|---|---|
| | | | In-Domain | OOD | |
| CIFAR-10 | $32 \times 32$ | 10 | CIFAR-10 training set (50,000 images) | CIFAR-100 training set (50,000 images) | In-Domain: CIFAR-10 test set (10,000 Images) OOD: TIM, LSUN, etc. |
| CIFAR-100 | $32 \times 32$ | 100 | CIFAR-100 training set (50,000 images) | CIFAR-10 training set (50,000 images) | In-Domain: CIFAR-10 test set (10,000 Images) OOD: TIM, LSUN, etc. |

CIFAR-100, we use CIFAR-100 training images as in-domain and CIFAR-10 training images as the OOD dataset. Here, we consider a DenseNet with depth $= 55$ and growth rate $= 12$ (Huang et al., 2017).

We present in-domain *misclassification detection* and *OOD detection* experiments to evaluate the performance of our models. For our *misclassification detection* experiments, we take the in-domain test set and attempt to distinguish the correctly classified examples from misclassified examples (see Table 2). The *OOD detection* experiments attempt to distinguish the in-domain test images from unknown OOD images. In Table 3, we present a smaller set of results where we consider *TinyImageNet (TIM)* as the OOD dataset and attempt to distinguish them from in-domain test data points. In Appendix A, we present an expanded version of this comparative table for a wide range of OOD examples. Note that, the OOD test images are selected from datasets different from the datasets used for training. The description of our training and test datasets are presented in Table 1. Please refer to Appendix C for further details.

**Evaluation of Predictive Uncertainty Estimation.** To evaluate the performance of our model for misclassification detection, we consider the misclassified examples as the positive class and correctly classified examples as the negative class. For the OOD detection task, we treat the OOD examples as the positive class and in-domain examples as a negative class. The detection performance for these tasks are measured using two metrics: *area under the receiver operating characteristic curve (AUROC)* and *area under the precision-recall curve (AUPR)* (Hendrycks & Gimpel, 2016). The AUROC can be interpreted as the probability of an OOD example to produce a higher detection score than an in-domain example (Davis & Goadrich, 2006). Hence, a higher AUROC is desirable, and an uninformative detector produces an AUROC $\approx 50\%$. The AUPR is more informative when the positive class and negative class have greatly differing base rates. It can take these base rates into account (Manning & Schütze, 1999).

| $\mathcal{D}_{in}^{test}$ | Methods | AUROC | | | | AUPR | | | | Acc.(%) |
|---|---|---|---|---|---|---|---|---|---|---|
| | | Max.P | Ent. | $\sum e^{z_c(x^*)}$ | D-Ent | Max.P | Ent. | $\sum e^{z_c(x^*)}$ | D-Ent | |
| CIFAR-10 | Baseline | 93.2 | 93.3 | - | - | 43.0 | 46.6 | - | - | **94.1** |
| | MCDP | **93.6** | **93.6** | - | - | 46.1 | 46.3 | - | - | **94.1** |
| | ODIN | 91.1 | - | - | - | 48.5 | - | - | - | **94.1** |
| | OE | 92.1 | 91.6 | - | - | 36.4 | 34.8 | - | - | 94.0 |
| | DPN$_{Dir}$ | 92.2 | 92.1 | - | 90.9 | **52.7** | 51.0 | - | 45.5 | 92.5 |
| | DPN$_{soft}(\beta:0.0,\sigma:0.0)$ | 91.7 | 91.3 | 90.1 | 90.2 | 37.0 | 35.7 | 32.2 | 32.5 | 94.0 |
| | DPN$_{soft}(\beta:0.0,\sigma:0.01)$ | 91.3 | 90.8 | 88.5 | 88.9 | 35.0 | 33.3 | 29.1 | 29.7 | 93.6 |
| | DPN$_{soft}(\beta:0.0,\sigma:0.05)$ | 93.4 | 93.2 | 91.4 | 91.8 | **44.7** | 42.5 | 35.1 | 36.4 | 93.7 |
| | DPN$_{soft}(\beta:0.5,\sigma:0.0)$ | 92.0 | 91.6 | 89.9 | 60.3 | 36.8 | 34.8 | 31.5 | 17.0 | **94.1** |
| | DPN$_{soft}(\beta:0.5,\sigma:0.01)$ | 91.3 | 90.8 | 87.7 | 61.9 | 37.1 | 35.3 | 28.9 | 17.0 | **94.1** |
| | DPN$_{soft}(\beta:0.5,\sigma:0.05)$ | 93.0 | 92.8 | 90.1 | 88.8 | 39.8 | 38.3 | 29.8 | 32.8 | **94.1** |
| CIFAR-100 | Baseline | 86.8 | 87.0 | - | - | 63.6 | 64.7 | - | - | 76.3 |
| | MCDP | **87.7** | **87.7** | - | - | 65.7 | **65.8** | - | - | **76.9** |
| | ODIN | 79.5 | - | - | - | 55.3 | - | - | - | 76.3 |
| | OE | 86.0 | 85.0 | - | - | 59.3 | 55.5 | - | - | 76.3 |
| | DPN$_{soft}(\beta:0.0,\sigma:0.0)$ | 86.5 | 85.4 | 74.4 | 75.0 | 60.7 | 57.1 | 42.6 | 41.8 | 75.7 |
| | DPN$_{soft}(\beta:0.0,\sigma:0.01)$ | 86.9 | 86.2 | 75.1 | 75.6 | 61.5 | 58.5 | 43.3 | 43.1 | 75.6 |
| | DPN$_{soft}(\beta:0.0,\sigma:0.05)$ | 87.3 | 86.7 | 76.7 | 76.4 | **64.6** | 61.7 | 45.8 | 43.3 | 75.7 |
| | DPN$_{soft}(\beta:0.5,\sigma:0.0)$ | 86.7 | 85.6 | 73.5 | 73.1 | 61.3 | 57.3 | 41.4 | 41.9 | 76.0 |
| | DPN$_{soft}(\beta:0.5,\sigma:0.01)$ | 86.5 | 85.7 | 74.1 | 74.6 | 61.0 | 57.5 | 41.4 | 43.2 | 76.2 |
| | DPN$_{soft}(\beta:0.5,\sigma:0.05)$ | **87.6** | 87.0 | 76.3 | 77.1 | **64.6** | 61.9 | 45.4 | 46.1 | 76.3 |

Table 2: Comparative results of misclassified image detection for CIFAR-10 and CIFAR-100.

**Hyper-parameters for Training Loss.** Unlike Liang et al. (2018); Lee et al. (2018a), and similar to Malinin & Gales (2018); Hendrycks et al. (2019), we *do not* require to tune any hyper-parameters during testing for different datasets. In other words, the OOD test examples remain unknown to our DPN classifiers, as in a real-world scenario. During training, we set $\lambda = 0.5$ (eq. 12), similar to Hendrycks et al. (2019). We train multiple DPN models for CIFAR-10 and CIFAR-100 classifiers using both positive and negative values for $\lambda_{out}$. We choose $\lambda_{in}$ and $\lambda_{out}$ as $\lambda_{in} = (1 - \beta)$ and $\lambda_{out} = (\frac{1}{\#class} - \beta)$. We train two different sets of DPN models using 0.0 (i.e positive $\lambda_{out}$) and $\beta = 0.5$ (i.e negative $\lambda_{out}$), denoted as DPN$_{soft}(\beta : 0.0)$ and DPN$_{soft}(\beta : 0.5)$, respectively.

**Detecting the source of uncertainty. [1] DPN model with $\lambda_{out} > 0$:** By choosing a positive value for $\lambda_{out}$, we enforce the network to produce *flat Dirichlet* distributions over the simplex for OOD examples. Hence, the D-Ent measure should produce higher scores for OOD examples and lower scores for in-domain examples. Therefore, given a test example with high scores for both entropy as well as D-Ent, it indicates *distributional uncertainty*. In contrast, if the test example achieves high entropy scores and lower D-Ent score, it indicates *data uncertainty*. Our DPN$_{soft}(\beta : 0.0)$ models achieve high AUROC and AUPR scores using D-Ent to detect OOD examples for both CIFAR-10 and CIFAR-100 (see Table 3) while achieves lower AUROC and AUPR scores for D-Ent while detecting misclasification for CIFAR-100 (see Table 2). However for CIFAR-

| $\mathcal{D}_{in}^{test}$ | $\mathcal{D}_{out}^{test}$ | Methods | AUROC | | | | AUPR | | | |
|---|---|---|---|---|---|---|---|---|---|---|
| | | | Max.P | Ent. | $\sum e^{z_c(x^*)}$ | D-Ent | Max.P | Ent. | $\sum e^{z_c(x^*)}$ | D-Ent |
| CIFAR-10 | TIM | Baseline | 88.8 | 89.4 | - | - | 85.1 | 86.7 | - | - |
| | | MCDP | 88.5 | 89.2 | - | - | 84.7 | 86.1 | - | - |
| | | ODIN | 94.4 | - | - | - | 93.8 | - | - | - |
| | | OE | 98.0 | 98.0 | - | - | 97.9 | 97.9 | - | - |
| | | $DPN_{Dir}$ | 94.3 | 94.3 | - | 94.6 | 94.0 | 94.0 | - | 94.2 |
| | | $DPN_{soft}(\beta:0.0,\sigma:0.0)$ | 97.6 | 97.7 | 97.6 | 97.6 | 97.5 | 97.6 | 97.5 | 97.5 |
| | | $DPN_{soft}(\beta:0.0,\sigma:0.01)$ | 98.5 | 98.5 | 98.4 | 98.5 | 98.4 | 98.5 | 98.3 | 98.4 |
| | | $DPN_{soft}(\beta:0.0,\sigma:0.05)$ | 97.1 | 97.4 | 97.6 | 97.7 | 97.2 | 97.5 | 97.7 | 97.8 |
| | | $DPN_{soft}(\beta:0.5,\sigma:0.0)$ | 98.7 | 98.8 | 96.7 | 6.8 | 98.6 | 98.7 | 92.8 | 32.5 |
| | | $DPN_{soft}(\beta:0.5,\sigma:0.01)$ | 99.0 | **99.1** | 96.3 | 6.8 | 98.5 | **98.9** | 90.7 | 32.2 |
| | | $DPN_{soft}(\beta:0.5,\sigma:0.05)$ | 97.9 | 98.2 | 98.2 | 30.9 | 97.9 | 98.2 | 98.1 | 54.1 |
| CIFAR-100 | TIM | Baseline | 74.9 | 76.3 | - | - | 71.1 | 73.1 | - | - |
| | | MCDP | 78.9 | 81.0 | - | - | 75.4 | 78.0 | - | - |
| | | ODIN | 83.8 | - | - | - | 81.4 | - | - | - |
| | | OE | 86.5 | 88.0 | - | - | 82.8 | 83.0 | - | - |
| | | $DPN_{soft}(\beta:0.0,\sigma:0.0)$ | 88.9 | 90.3 | 91.1 | 90.7 | 86.1 | 86.4 | 85.4 | 83.6 |
| | | $DPN_{soft}(\beta:0.0,\sigma:0.01)$ | 96.5 | 97.4 | 98.8 | 98.0 | 97.1 | 97.8 | 98.9 | 95.2 |
| | | $DPN_{soft}(\beta:0.0,\sigma:0.05)$ | 95.8 | 96.7 | 98.0 | 96.7 | 96.4 | 97.2 | 98.2 | 92.9 |
| | | $DPN_{soft}(\beta:0.5,\sigma:0.0)$ | 98.9 | 99.1 | 99.5 | 5.0 | 98.9 | 99.0 | 99.5 | 31.8 |
| | | $DPN_{soft}(\beta:0.5,\sigma:0.01)$ | 99.1 | 99.3 | **99.7** | 5.9 | 99.2 | 99.3 | **99.7** | 32.4 |
| | | $DPN_{soft}(\beta:0.5,\sigma:0.05)$ | 95.8 | 96.9 | 98.8 | 41.8 | 96.4 | 97.4 | 98.9 | 59.8 |

Table 3: Comparative results of OOD example detection for CIFAR-10 and CIFAR-100. Expanded version of this table along with a wide range of OOD datasets are provided in Appendix A.

10, our $DPN_{soft}(\beta:0.0)$ models fail to produce low AUROC and AUPR scores under D-Ent measure for misclassification detection. This indicates that these models achieve high D-Ent scores for both misclassified and OOD examples and make it difficult to distinguish distributional uncertainty from data uncertainty for CIFAR-10.

Note that, similar to our $DPN_{soft}(\beta:0.0)$, $DPN_{Dir}$ (Malinin & Gales, 2018) also produces high AUPR and AUROC scores under D-Ent measure for for both misclassified examples and OOD examples for CIFAR-10 model (Table 2 and 3). This indicates that both $DPN_{Dir}$ and our $DPN_{soft}(\beta:0.0)$ models may be ineffective to detect distributional uncertainty for CIFAR-10. In contrast, our $DPN_{soft}(\beta:0.5)$ models are always found to be effective to identify the source of uncertainties.

**[2] DPN model with $\lambda_{out} < 0$:** In contrast, if we choose a negative value for $\lambda_{out}$, our DPN network produces very sharp Dirichlet distributions across the edge of the simplex for OOD examples. Hence, it produces very low D-Ent scores. Notably, these distributions are even sharper than the Dirichlet distributions obtained for confidently predicted examples. Hence, for a test example, we can detect *distributional uncertainty* if it achieves high entropy and low D-Ent scores. As we can see, our $DPN_{soft}(\beta:0.5)$ models produce very low AUROC and AUPR scores for D-Ent measure to detect OOD examples (Table 3) while achieve relatively higher AUROC and AUPR scores for detecting misclassified examples (Table 2).

**Data Augmentation with white-noise for Training.** After training our DPN models using clean training images, we also fine-tune them using noisy OOD training images for a few epochs. We add minor Gaussian noise sampled from isotropic Gaussian distributions $\mathcal{N}(0, \sigma^2 I)$ with our OOD training images. Here, the idea is to add *minor perturbation* without distorting the features of the OOD training images. It further exposes the network into the out of distribution space. Note that, the DPN models are fine-tuned *only* using OOD training images. The OOD test examples remain *unknown*. In practice, this technique may often further improve the OOD detection performance of our $DPN_{soft}$ models as we choose $\sigma = 0.01$ (see Table 3). However, for larger perturbation, performance degrades for the $DPN_{soft}(\sigma = 0.05)$.

**Comparative Study.** In Table 2 and 3, we compare the performance of our approach with several baselines, such as standard DNN (Hendrycks & Gimpel (2016)), MCDP (Gal & Ghahramani, 2016), $DPN_{Dir}$ (Malinin & Gales, 2018), ODIN (Liang et al., 2018), and OE (Hendrycks et al., 2019). We use the same architecture as our $DPN_{soft}$ models for all the competitive models. OE models are trained using the set of in-domain and OOD training images with their proposed loss function (Hendrycks et al., 2019). Note that, since non-DPN methods do not explicitly model the logit outputs, the D-Ent or $\sum_{c=1}^{K} e^{z_c(x^*)}$ measures are not meaningful (Malinin & Gales, 2018).

We also compare our results with the existing DPN framework, namely $DPN_{Dir}$ Malinin & Gales (2018) for CIFAR-10. Note that, the $DPN_{Dir}$ framework fails to scale for the CIFAR-100 dataset that consists of 100 classes. Due to the unavailability of codes and results (under the same settings), we could not compare our model with Malinin & Gales (2019). Overall, our $DPN_{soft}$ models significantly improved the performance of the DPN framework and consistently out-performed the existing OOD detection models. In addition, it is able to distinguish distributional uncertainty from other uncertainty types.

## 5 CONCLUSION

In this paper, we propose a novel framework to improve the training efficiency of DPN models for challenging classification tasks with large number of classes. We also propose a novel regularization term that allows controls the sharpness of the Dirichlet distributions. We show that the proposed regularizer can be easily integrated

with the standard cross-entropy loss function. Our experiments on synthetic and real datasets demonstrate that our proposed framework can efficiently distinguish the distributional uncertainty from other uncertainty types. We demonstrate that the OOD detection performance of our DPN models can be often improved by training with noisy OOD examples. Experiments show that our proposed approach significantly improves DPN frameworks, and outperforms the existing OOD detectors on both CIFAR-10 and CIFAR-100 datasets.

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

# A    EXPANDED RESULTS

Table 4: Results of OOD image detection for CIFAR-10. Description of these OOD datasets are provided in Appendix C.2.

| $\mathcal{D}_{out}^{test}$ | Methods | AUROC | | | | AUPR | | | |
|---|---|---|---|---|---|---|---|---|---|
| | | Max.P | Ent. | $\sum e^{z_c(\boldsymbol{x}^*)}$ | D-Ent | Max.P | Ent. | $\sum e^{z_c(\boldsymbol{x}^*)}$ | D-Ent |
| Gaussian | Baseline | 73.0 | 73.3 | - | - | 70.4 | 71.3 | - | - |
| | MCDP | 74.0 | 74.0 | - | - | 70.9 | 71.2 | - | - |
| | ODIN | 72.6 | - | - | - | 70.1 | - | - | - |
| | OE | 81.9 | 82.0 | - | - | 82.5 | 82.6 | - | - |
| | $\text{DPN}_{soft}(\beta:0.0,\sigma:0.0)$ | 82.3 | 82.4 | 82.7 | 82.7 | 83.9 | 84.1 | 84.0 | 84.0 |
| | $\text{DPN}_{soft}(\beta:0.0,\sigma:0.01)$ | 96.0 | 96.2 | **96.4** | **96.4** | 96.3 | 96.7 | **96.7** | **96.7** |
| | $\text{DPN}_{soft}(\beta:0.0,\sigma:0.05)$ | 89.2 | 89.6 | 90.8 | 90.7 | 91.6 | 92.0 | 93.0 | 92.9 |
| | $\text{DPN}_{soft}(\beta:0.5,\sigma:0.0)$ | 82.9 | 83.0 | 83.3 | 37.3 | 83.8 | 83.9 | 82.6 | 48.7 |
| | $\text{DPN}_{soft}(\beta:0.5,\sigma:0.01)$ | 95.8 | 96.0 | 94.6 | 13.0 | 96.4 | 96.7 | 90.9 | 35.1 |
| | $\text{DPN}_{soft}(\beta:0.5,\sigma:0.05)$ | 90.2 | 90.6 | 92.4 | 28.1 | 92.4 | 92.8 | 94.0 | 47.3 |
| TIM | Baseline | 88.8 | 89.4 | - | - | 85.1 | 86.7 | - | - |
| | MCDP | 88.5 | 89.2 | - | - | 84.7 | 86.1 | - | - |
| | ODIN | 94.4 | - | - | - | 93.8 | - | - | - |
| | OE | 98.0 | 98.0 | - | - | 97.9 | 97.9 | - | - |
| | $\text{DPN}_{Dir}$ | 94.3 | 94.3 | - | 94.6 | 94.0 | 94.0 | | 94.2 |
| | $\text{DPN}_{soft}(\beta:0.0,\sigma:0.0)$ | 97.6 | 97.7 | 97.6 | 97.6 | 97.5 | 97.6 | 97.5 | 97.5 |
| | $\text{DPN}_{soft}(\beta:0.0,\sigma:0.01)$ | 98.5 | 98.5 | 98.4 | 98.5 | 98.4 | 98.5 | 98.3 | 98.4 |
| | $\text{DPN}_{soft}(\beta:0.0,\sigma:0.05)$ | 97.1 | 97.4 | 97.6 | 97.7 | 97.2 | 97.5 | 97.7 | 97.8 |
| | $\text{DPN}_{soft}(\beta:0.5,\sigma:0.0)$ | 98.7 | 98.8 | 96.7 | 6.8 | 98.6 | 98.7 | 92.8 | 32.5 |
| | $\text{DPN}_{soft}(\beta:0.5,\sigma:0.01)$ | 99.0 | **99.1** | 96.3 | 6.8 | 98.5 | **98.9** | 90.7 | 32.2 |
| | $\text{DPN}_{soft}(\beta:0.5,\sigma:0.05)$ | 97.9 | 98.2 | 98.2 | 30.9 | 97.9 | 98.2 | 98.1 | 54.1 |
| LSUN | Baseline | 90.2 | 91.0 | - | - | 86.6 | 88.6 | - | - |
| | MCDP | 90.3 | 91.1 | - | - | 86.8 | 88.9 | - | - |
| | ODIN | 96.6 | - | - | - | 96.2 | - | - | - |
| | OE | 97.9 | 98.0 | - | - | 97.6 | 97.7 | - | - |
| | $\text{DPN}_{Dir}$ | 94.4 | 94.4 | | 94.6 | 93.3 | 93.4 | | 93.3 |
| | $\text{DPN}_{soft}(\beta:0.0,\sigma:0.0)$ | 97.7 | 97.8 | 97.7 | 97.7 | 97.3 | 97.4 | 97.3 | 97.3 |
| | $\text{DPN}_{soft}(\beta:0.0,\sigma:0.01)$ | 98.7 | 98.7 | 98.6 | 98.6 | 98.5 | 98.5 | 98.3 | 98.3 |
| | $\text{DPN}_{soft}(\beta:0.0,\sigma:0.05)$ | 97.3 | 97.7 | 97.9 | 98.0 | 97.2 | 97.6 | 97.7 | 97.8 |
| | $\text{DPN}_{soft}(\beta:0.5,\sigma:0.0)$ | 98.8 | 98.8 | 97.1 | 6.2 | 98.6 | 98.6 | 93.8 | 32.4 |
| | $\text{DPN}_{soft}(\beta:0.5,\sigma:0.01)$ | **99.2** | **99.2** | 96.7 | 5.5 | 98.7 | **98.9** | 91.8 | 31.8 |
| | $\text{DPN}_{soft}(\beta:0.5,\sigma:0.05)$ | 97.9 | 98.2 | 98.4 | 33.0 | 97.7 | 98.1 | 98.3 | 56.0 |
| Places365 | Baseline | 89.4 | 90.0 | - | - | 95.6 | 96.2 | - | - |
| | MCDP | 89.3 | 90.2 | - | - | 95.6 | 96.2 | - | - |
| | ODIN | 95.4 | - | - | - | 98.5 | - | - | - |
| | OE | 97.9 | 98.0 | - | - | 99.4 | 99.4 | - | - |
| | $\text{DPN}_{soft}(\beta:0.0,\sigma:0.0)$ | 97.7 | 97.8 | 97.7 | 97.7 | 99.3 | 99.3 | 99.3 | 99.3 |
| | $\text{DPN}_{soft}(\beta:0.0,\sigma:0.01)$ | 98.8 | 98.8 | 98.7 | 98.7 | 99.6 | 99.6 | 99.6 | 99.6 |
| | $\text{DPN}_{soft}(\beta:0.0,\sigma:0.05)$ | 97.3 | 97.6 | 97.8 | 97.8 | 99.2 | 99.3 | 99.4 | 99.4 |
| | $\text{DPN}_{soft}(\beta:0.5,\sigma:0.0)$ | 98.7 | 98.7 | 96.7 | 6.7 | 99.6 | 99.6 | 97.9 | 60.7 |
| | $\text{DPN}_{soft}(\beta:0.5,\sigma:0.01)$ | 99.2 | **99.3** | 96.5 | 5.4 | 99.6 | **99.7** | 97.3 | 59.5 |
| | $\text{DPN}_{soft}(\beta:0.5,\sigma:0.05)$ | 97.9 | 98.2 | 98.3 | 29.5 | 99.4 | 99.5 | 99.5 | 77.1 |
| Textures | Baseline | 88.6 | 89.0 | - | - | 74.9 | 77.0 | - | - |
| | MCDP | 87.6 | 88.0 | - | - | 73.7 | 75.8 | - | - |
| | ODIN | 94.9 | - | - | - | 90.5 | - | - | - |
| | OE | 99.2 | 99.7 | - | - | 98.6 | 98.5 | - | - |
| | $\text{DPN}_{soft}(\beta:0.0,\sigma:0.0)$ | 99.5 | 99.5 | 99.5 | 99.5 | 98.8 | 99.0 | 99.0 | 99.0 |
| | $\text{DPN}_{soft}(\beta:0.0,\sigma:0.01)$ | 99.5 | 99.5 | 99.3 | 99.4 | 98.9 | 98.9 | 98.6 | 98.7 |
| | $\text{DPN}_{soft}(\beta:0.0,\sigma:0.05)$ | 99.4 | 99.5 | 99.5 | **99.6** | 99.0 | 99.1 | 99.2 | **99.3** |
| | $\text{DPN}_{soft}(\beta:0.5,\sigma:0.0)$ | 99.5 | 99.5 | 96.7 | 4.6 | 99.1 | 99.1 | 85.6 | 21.3 |
| | $\text{DPN}_{soft}(\beta:0.5,\sigma:0.01)$ | 99.4 | 99.5 | 96.5 | 5.5 | 98.0 | 98.8 | 84.3 | 21.4 |
| | $\text{DPN}_{soft}(\beta:0.5,\sigma:0.05)$ | 99.1 | 99.3 | 98.7 | 24.7 | 98.3 | 98.7 | 97.3 | 38.3 |

# B    EXPERIMENTAL DETAILS ON SYNTHETIC DATASETS

The three classes of our synthetic dataset are constructed by sampling from three different isotropic Gaussian distributions with means of $(-4, 0)$, $(4, 0)$ and $(0, 5)$ and isotropic variances of $\sigma = 4$. We sample 200 training data points from each distribution for each class. We also sample 600 OOD training examples from an uniform distribution of $\mathcal{U}([-15, 15], [-13, 17])$.

Table 5: Results of OOD image detection for CIFAR-100. Description of these OOD datasets are provided in Appendix C.2.

| $\mathcal{D}_{out}^{test}$ | Methods | AUROC | | | | AUPR | | | |
|---|---|---|---|---|---|---|---|---|---|
| | | Max.P | Ent. | $\sum e^{z_c(\boldsymbol{x}^*)}$ | D-Ent | Max.P | Ent. | $\sum e^{z_c(\boldsymbol{x}^*)}$ | D-Ent |
| Gaussian | Baseline | 73.2 | 74.0 | - | - | 67.7 | 68.5 | - | - |
| | MCDP | 74.4 | 74.7 | - | - | 68.9 | 69.3 | - | - |
| | ODIN | 80.4 | - | - | - | 76.2 | - | - | - |
| | OE | 72.9 | 74.0 | - | - | 64.6 | 64.9 | - | - |
| | $\text{DPN}_{soft}(\beta:0.0,\sigma:0.0)$ | 71.7 | 72.6 | 71.9 | 71.9 | 63.6 | 63.9 | 61.7 | 61.5 |
| | $\text{DPN}_{soft}(\beta:0.0,\sigma:0.01)$ | 94.4 | 95.2 | 97.6 | 96.4 | 95.3 | 96.0 | 97.9 | 94.0 |
| | $\text{DPN}_{soft}(\beta:0.0,\sigma:0.05)$ | 96.1 | 96.7 | 98.7 | 97.2 | 97.0 | 97.5 | 98.9 | 93.5 |
| | $\text{DPN}_{soft}(\beta:0.5,\sigma:0.0)$ | 91.9 | 92.8 | 96.3 | 37.6 | 91.8 | 92.4 | 95.8 | 50.8 |
| | $\text{DPN}_{soft}(\beta:0.5,\sigma:0.01)$ | 98.3 | 98.6 | **99.6** | 3.4 | 98.6 | 98.8 | **99.7** | 31.0 |
| | $\text{DPN}_{soft}(\beta:0.5,\sigma:0.05)$ | 96.9 | 97.4 | **99.6** | 9.3 | 97.7 | 98.1 | **99.7** | 33.6 |
| TIM | Baseline | 74.9 | 76.3 | - | - | 71.1 | 73.1 | - | - |
| | MCDP | 78.9 | 81.0 | - | - | 75.4 | 78.0 | - | - |
| | ODIN | 83.8 | - | - | - | 81.4 | - | - | - |
| | OE | 86.5 | 88.0 | - | - | 82.8 | 83.0 | - | - |
| | $\text{DPN}_{soft}(\beta:0.0,\sigma:0.0)$ | 88.9 | 90.3 | 91.1 | 90.7 | 86.1 | 86.4 | 85.4 | 83.6 |
| | $\text{DPN}_{soft}(\beta:0.0,\sigma:0.01)$ | 96.5 | 97.4 | 98.8 | 98.0 | 97.1 | 97.8 | **98.9** | 95.2 |
| | $\text{DPN}_{soft}(\beta:0.0,\sigma:0.05)$ | 95.8 | 96.7 | 98.0 | 96.7 | 96.4 | 97.2 | 98.2 | 92.9 |
| | $\text{DPN}_{soft}(\beta:0.5,\sigma:0.0)$ | 98.9 | 99.1 | 99.5 | 5.0 | 98.9 | 99.0 | 99.5 | 31.8 |
| | $\text{DPN}_{soft}(\beta:0.5,\sigma:0.01)$ | 99.1 | 99.3 | **99.7** | 5.9 | 99.2 | 99.3 | **99.7** | 32.4 |
| | $\text{DPN}_{soft}(\beta:0.5,\sigma:0.05)$ | 95.8 | 96.9 | 98.8 | 41.8 | 96.4 | 97.4 | 98.9 | 59.8 |
| LSUN | Baseline | 78.9 | 80.4 | - | - | 74.2 | 75.9 | - | - |
| | MCDP | 83.2 | 85.4 | - | - | 79.0 | 81.5 | - | - |
| | ODIN | 87.8 | - | - | - | 84.6 | - | - | - |
| | OE | 90.6 | 91.6 | - | - | 86.5 | 86.1 | - | - |
| | $\text{DPN}_{soft}(\beta:0.0,\sigma:0.0)$ | 91.6 | 92.6 | 93.3 | 92.6 | 88.3 | 88.3 | 87.3 | 85.4 |
| | $\text{DPN}_{soft}(\beta:0.0,\sigma:0.01)$ | 98.9 | 99.2 | 99.7 | 98.9 | 99.0 | 99.3 | 99.7 | 96.1 |
| | $\text{DPN}_{soft}(\beta:0.0,\sigma:0.05)$ | 96.0 | 96.8 | 98.2 | 96.8 | 96.6 | 97.2 | 98.4 | 93.0 |
| | $\text{DPN}_{soft}(\beta:0.5,\sigma:0.0)$ | 99.7 | 99.7 | **99.9** | 1.4 | 99.6 | 99.7 | 99.8 | 30.8 |
| | $\text{DPN}_{soft}(\beta:0.5,\sigma:0.01)$ | 99.7 | 99.7 | **99.9** | 2.8 | 99.7 | 99.7 | **99.9** | 31.1 |
| | $\text{DPN}_{soft}(\beta:0.5,\sigma:0.05)$ | 96.8 | 97.5 | 99.1 | 37.1 | 97.3 | 98.0 | 99.2 | 56.2 |
| Places365 | Baseline | 76.6 | 78.1 | - | - | 90.2 | 91.0 | - | - |
| | MCDP | 80.9 | 83.1 | - | - | 92.2 | 93.3 | - | - |
| | ODIN | 86.4 | - | - | - | 94.7 | - | - | - |
| | OE | 88.8 | 90.1 | - | - | 95.1 | 95.0 | | - |
| | $\text{DPN}_{soft}(\beta:0.0,\sigma:0.0)$ | 90.5 | 91.3 | 92.6 | 92.0 | 96.0 | 96.0 | 95.9 | 94.9 |
| | $\text{DPN}_{soft}(\beta:0.0,\sigma:0.01)$ | 97.8 | 98.4 | 99.3 | 98.5 | 99.4 | 99.6 | 99.8 | 98.6 |
| | $\text{DPN}_{soft}(\beta:0.0,\sigma:0.05)$ | 95.9 | 96.7 | 98.1 | 96.7 | 98.9 | 99.1 | 99.5 | 97.6 |
| | $\text{DPN}_{soft}(\beta:0.5,\sigma:0.0)$ | 99.3 | 99.4 | 99.7 | 3.4 | 99.8 | 99.8 | **99.9** | 59.1 |
| | $\text{DPN}_{soft}(\beta:0.5,\sigma:0.01)$ | 99.4 | 99.5 | **99.8** | 4.8 | 99.8 | **99.9** | **99.9** | 60.3 |
| | $\text{DPN}_{soft}(\beta:0.5,\sigma:0.05)$ | 96.0 | 96.9 | 98.8 | 40.0 | 98.9 | 99.2 | 99.7 | 80.9 |
| Textures | Baseline | 60.0 | 60.2 | - | - | 43.4 | 43.4 | - | - |
| | MCDP | 64.0 | 64.3 | - | - | 46.0 | 45.6 | - | - |
| | ODIN | 63.4 | - | - | - | 48.9 | - | - | - |
| | OE | 73.6 | 74.8 | - | - | 58.3 | 58.4 | - | - |
| | $\text{DPN}_{soft}(\beta:0.0,\sigma:0.0)$ | 80.5 | 81.8 | 85.0 | 83.7 | 66.6 | 66.9 | 71.3 | 67.2 |
| | $\text{DPN}_{soft}(\beta:0.0,\sigma:0.01)$ | 84.3 | 86 | 92.9 | 90.9 | 81.3 | 83.3 | 90.9 | 84.1 |
| | $\text{DPN}_{soft}(\beta:0.0,\sigma:0.05)$ | 89.2 | 90.7 | 94.5 | 92.8 | 86.2 | 88.3 | 93.1 | 84.5 |
| | $\text{DPN}_{soft}(\beta:0.5,\sigma:0.0)$ | 89.7 | 91.2 | 96.2 | 10.6 | 86.1 | 87.8 | 95.1 | 22.4 |
| | $\text{DPN}_{soft}(\beta:0.5,\sigma:0.01)$ | 92.2 | 93.6 | **97.4** | 8.9 | 89.9 | 91.6 | **96.9** | 22.1 |
| | $\text{DPN}_{soft}(\beta:0.5,\sigma:0.05)$ | 86.7 | 88.9 | 96.6 | 29.6 | 83.7 | 86.8 | 96.1 | 35.1 |

We train a neural network with 2 hidden layers with 50 nodes each and $relu$ activation function. The network is trained for $2,500$ epochs using stochastic gradient descent (SGD) optimization with a constant learning rate of 0.01.

## C  EXPERIMENTAL DETAILS ON CIFAR-10 AND CIFAR-100

### C.1  TRAINING DETAILS

For our experiments on CIFAR-10, we train a VGG-16 model with CIFAR-10 as the in-domain and CIFAR-100 as the OOD training data (Simonyan & Zisserman (2014b)). For CIFAR-100, we train a DenseNet with

Table 6: Results for *misclassification detection* for TinyImageNet.

| $\mathcal{D}_{in}^{test}$ | Methods | AUROC | | | | AUPR | | | | Acc.(%) |
|---|---|---|---|---|---|---|---|---|---|---|
| | | Max.P | Ent. | $\sum e^{z_c(\boldsymbol{x}^*)}$ | D-Ent | Max.P | Ent. | $\sum e^{z_c(\boldsymbol{x}^*)}$ | D-Ent | |
| TinyImageNet | Baseline | 85.4 | 85.2 | - | - | 78.3 | 77.8 | - | - | **57.8** |
| | MCDP | 85.8 | 85.8 | - | - | 77.9 | 77.9 | - | - | **59.0** |
| | OE | 85.2 | 85.4 | - | - | 77.8 | 77.9 | - | - | 57.7 |
| | $\text{DPN}_{soft}(\beta:0.0,\sigma:0.0)$ | 85.6 | 85.6 | 81.7 | 80.0 | 78.8 | 78.6 | 73.5 | 73.5 | 57.8 |
| | $\text{DPN}_{soft}(\beta:0.0,\sigma:0.01)$ | 85.7 | 85.8 | 81.8 | 80.2 | 79.0 | 78.7 | 73.6 | 73.6 | 57.8 |
| | $\text{DPN}_{soft}(\beta:0.0,\sigma:0.05)$ | 85.4 | 85.6 | 81.8 | 80.3 | 78.4 | 78.5 | 73.6 | 73.6 | 57.8 |
| | $\text{DPN}_{soft}(\beta:0.5,\sigma:0.0)$ | 85.9 | **86.3** | 81.8 | 77.6 | 79.7 | **80.1** | 73.6 | 72.7 | 57.5 |
| | $\text{DPN}_{soft}(\beta:0.5,\sigma:0.01)$ | 85.8 | 86.2 | 81.9 | 77.5 | 79.5 | **80.1** | 73.7 | 72.8 | 57.5 |
| | $\text{DPN}_{soft}(\beta:0.5,\sigma:0.05)$ | 85.7 | 86.1 | 81.8 | 77.9 | 79.2 | 80.0 | 73.7 | 73.1 | 57.6 |

Table 7: Results of OOD image detection for TinyImageNet classifiers. Description of these OOD datasets are provided in Appendix C.2.

| $\mathcal{D}_{out}^{test}$ | Methods | AUROC | | | | AUPR | | | |
|---|---|---|---|---|---|---|---|---|---|
| | | Max.P | Ent. | $\sum e^{z_c(\boldsymbol{x}^*)}$ | D-Ent | Max.P | Ent. | $\sum e^{z_c(\boldsymbol{x}^*)}$ | D-Ent |
| STL-10 | Baseline | 62.9 | 61.7 | - | - | 51.4 | 50.0 | - | - |
| | MCDP | 63.8 | 63.6 | - | - | 52.2 | 51.8 | - | - |
| | OE | 98.9 | **99.0** | - | - | 99.0 | 99.1 | - | - |
| | $\text{DPN}_{soft}(\beta:0.0,\sigma:0.0)$ | 98.4 | 98.5 | 98.9 | 98.8 | 98.6 | 98.8 | 99.1 | 99.0 |
| | $\text{DPN}_{soft}(\beta:0.0,\sigma:0.01)$ | 98.1 | 98.5 | 98.6 | 98.6 | 98.4 | 98.6 | 98.9 | 98.8 |
| | $\text{DPN}_{soft}(\beta:0.0,\sigma:0.05)$ | 97.0 | 98.1 | 97.9 | 97.9 | 97.5 | 97.7 | 98.4 | 98.3 |
| | $\text{DPN}_{soft}(\beta:0.5,\sigma:0.0)$ | 98.6 | 98.7 | **99.0** | 4.3 | 98.8 | 98.9 | **99.2** | 28.3 |
| | $\text{DPN}_{soft}(\beta:0.5,\sigma:0.01)$ | 98.3 | 98.4 | 98.8 | 5.2 | 98.5 | 98.7 | 99.0 | 28.8 |
| | $\text{DPN}_{soft}(\beta:0.5,\sigma:0.05)$ | 97.5 | 97.7 | 98.4 | 7.6 | 97.8 | 98.0 | 98.6 | 30.3 |
| LSUN | Baseline | 64.9 | 65.6 | - | - | 59.7 | 59.5 | - | - |
| | MCDP | 65.9 | 66.1 | - | - | 60.3 | 60.2 | - | - |
| | OE | 98.2 | 98.5 | - | - | 98.7 | 99.0 | - | - |
| | $\text{DPN}_{soft}(\beta:0.0,\sigma:0.0)$ | 97.8 | 98.2 | 98.9 | 98.8 | 98.4 | 98.7 | 99.2 | 99.2 |
| | $\text{DPN}_{soft}(\beta:0.0,\sigma:0.01)$ | 97.3 | 97.7 | 98.7 | 98.6 | 98.1 | 98.4 | 99.0 | 99.0 |
| | $\text{DPN}_{soft}(\beta:0.0,\sigma:0.05)$ | 95.2 | 95.9 | 97.7 | 97.5 | 96.5 | 97.0 | 98.3 | 98.1 |
| | $\text{DPN}_{soft}(\beta:0.5,\sigma:0.0)$ | 98.2 | 97.6 | **99.1** | 6.8 | 99.1 | 98.9 | **99.4** | 35.0 |
| | $\text{DPN}_{soft}(\beta:0.5,\sigma:0.01)$ | 97.6 | 98.0 | 98.8 | 9.1 | 98.3 | 98.6 | 99.1 | 36.8 |
| | $\text{DPN}_{soft}(\beta:0.5,\sigma:0.05)$ | 96.1 | 96.6 | 98.0 | 14.4 | 97.1 | 97.1 | 98.6 | 40.8 |
| Places365 | Baseline | 67.8 | 68.9 | - | - | 85.8 | 86.0 | - | - |
| | MCDP | 68.5 | 68.9 | - | - | 86.1 | 86.2 | - | - |
| | OE | 95.8 | 96.5 | - | - | 98.9 | 99.1 | - | - |
| | $\text{DPN}_{soft}(\beta:0.0,\sigma:0.0)$ | 95.0 | 95.8 | 97.3 | 97.1 | 98.7 | 98.9 | 99.3 | 99.3 |
| | $\text{DPN}_{soft}(\beta:0.0,\sigma:0.01)$ | 94.1 | 94.9 | 96.7 | 96.5 | 98.5 | 98.7 | 99.1 | 99.1 |
| | $\text{DPN}_{soft}(\beta:0.0,\sigma:0.05)$ | 90.5 | 91.8 | 94.4 | 94.1 | 97.4 | 97.4 | 98.5 | 98.5 |
| | $\text{DPN}_{soft}(\beta:0.5,\sigma:0.0)$ | 95.6 | 96.2 | **97.6** | 17.0 | 98.9 | 99.0 | **99.4** | 69.6 |
| | $\text{DPN}_{soft}(\beta:0.5,\sigma:0.01)$ | 94.4 | 95.1 | 96.8 | 21.3 | 98.6 | 98.7 | 99.2 | 72.1 |
| | $\text{DPN}_{soft}(\beta:0.5,\sigma:0.05)$ | 91.6 | 92.6 | 95.0 | 30.8 | 97.8 | 98.0 | 98.7 | 77.0 |
| Textures | Baseline | 68.8 | 70.9 | - | - | 51.0 | 53.3 | - | - |
| | MCDP | 69.1 | 69.7 | - | - | 51.0 | 51.6 | - | - |
| | OE | 87.5 | 89.1 | - | - | 85.6 | 87.5 | - | - |
| | $\text{DPN}_{soft}(\beta:0.0,\sigma:0.0)$ | 85.1 | 86.8 | 88.4 | **89.0** | 82.6 | 84.7 | 86.0 | **86.6** |
| | $\text{DPN}_{soft}(\beta:0.0,\sigma:0.01)$ | 83.9 | 85.8 | 87.6 | 88.2 | 81.1 | 83.3 | 85.0 | 85.6 |
| | $\text{DPN}_{soft}(\beta:0.0,\sigma:0.05)$ | 81.4 | 83.5 | 85.6 | 86.2 | 77.3 | 79.9 | 81.9 | 82.6 |
| | $\text{DPN}_{soft}(\beta:0.5,\sigma:0.0)$ | 85.7 | 87.0 | 88.1 | 45.0 | 83.3 | 84.8 | 85.8 | 46.6 |
| | $\text{DPN}_{soft}(\beta:0.5,\sigma:0.01)$ | 84.5 | 85.8 | 87.1 | 47.5 | 81.6 | 83.3 | 84.4 | 48.4 |
| | $\text{DPN}_{soft}(\beta:0.5,\sigma:0.05)$ | 82.3 | 83.8 | 85.3 | 53.0 | 78.5 | 80.4 | 81.8 | 52.0 |

depth = 55, growth rate = 12 and CIFAR-100 as the in-domain and CIFAR-10 as the OOD training data (Huang et al. (2017)). We trained multiple $\text{DPN}_{soft}$ models using our proposed loss functions with different hyper-parameters.

For CIFAR-10, we use the VGG-16 network. Here, we use CIFAR-10 training images ($50,000$ images) as our in-domain training data and CIFAR-100 training images ($50,000$ images) as our OOD training data.

For CIFAR-100, Densenet(55, 12) is trained using the same setup as proposed by Huang et al. (2017). Here, we use CIFAR-100 training images ($50,000$ images) as our in-domain training data and CIFAR-10 training images ($50,000$ images) as our OOD training data.

For TinyImageNet (TIM), we use the VGG-16 network. Here, we use TIM training images (100,000 images) as our in-domain training data and ImageNet-25K images ($25,000$ images) as our OOD training data. ImageNet-25K is obtained by randomly selecting $25,000$ images from the ImageNet dataset (Deng et al., 2009).

After training the models with clean in-domain and OOD images, we further fine-tune the models using *noisy OOD training images* for 50 epochs with the learning rate of 0.0001. The noises are chosen form an isotropic Gaussian distribution, $\mathcal{N}(0, \sigma^2 I)$. We have experimented with three different values of $\sigma$ as $\{0.0, 0.01, 0.05\}$ to introduce different level of noises.

Table 8: Details of training and test datasets

| Classification Task | Input Shape | #Classes | Details of Training data | | Details of Test data |
|---|---|---|---|---|---|
| | | | In-Domain | OOD | |
| CIFAR-10 | $32 \times 32$ | 10 | CIFAR-10 training set (50,000 images) | CIFAR-100 training set (50,000 images) | In-Domain: CIFAR-10 test set (10,000 Images) OOD: TIM, LSUN, etc. |
| CIFAR-100 | $32 \times 32$ | 100 | CIFAR-100 training set (50,000 images) | CIFAR-10 training set (50,000 images) | In-Domain: CIFAR-100 test set (10,000 Images) OOD: TIM, LSUN, etc. |
| TinyImageNet (TIM) | $64 \times 64$ | 200 | TinyImageNet training set (100,000 images) | ImageNet-25K (25,000 randomly sampled images from ImageNet) | In-Domain: TIM test set (10,000 Images) OOD: STL-10, LSUN, etc. |

## C.2 OOD Test Datasets

We use a wide range of OOD dataset to evaluate the performance of our proposed OOD detection models. For CIFAR-10 and CIFAR-100 classifiers, these input test images are resized to $32 \times 32$, while for TinyImageNet classifiers, we resize them to $64 \times 64$. In this task, we attempt to distinguish the unknown OOD images from the corresponding in-domain test images from different classification tasks. We compute different uncertainty measures for these images for this purpose. For our evaluations, we use the following OOD images as described in the following.

1. **TinyImageNet (TIM)** (Li et al. (2017)). This is a subset of Imagenet dataset. We use the validation set, that contains $10,000$ test images from 200 different image classes for our evaluation during test time. This dataset is used as an OOD test dataset only for CIFAR-10 and CIFAR-100 classifiers. Note that, for TinyImageNet classifiers, this is the in-domain test set.

2. **LSUN** (Yu et al. (2015)). The Large-scale Scene UNderstanding dataset (LSUN) contains images of 10 different scene categories. We use its validation set, containing $10,000$ images, as an unknown OOD test set.

3. **Places 365** (Zhou et al. (2017)). The validation set of this dataset consists of 36500 images of 365 scene categories.

4. **Textures** (Cimpoi et al. (2014)) contains 5640 textural images in the wild belonging to 47 categories.

5. **STL-10** contains $8,000$ images of natural images from 10 different classes (Coates et al., 2011).

6. **Gaussian Noise.** This is an artificially generated dataset obtained by adding Gaussian noise to the in-domain test images. The Gaussian noises are sampled from an isotropic Gaussian distribution, $\mathcal{N}(0, \sigma^2 I)$ with $\sigma = 0.05$.

## C.3 Details of Competitive Systems

We compare the performance of our models with standard DNN as baseline model (Hendrycks & Gimpel (2016)), the Bayesian framework, monti-carlo dropout (MCDP) (Gal & Ghahramani (2016)), DPN$_{Dir}$ using the loss function proposed by Malinin & Gales (2018), non-Bayesian frameworks such as ODIN (Liang et al. (2018)) and outlier exposure (OE) by Hendrycks et al. (2019). We use the same architecture as DPN$_{softmax}$ for the competitive models. For DPN$_{Dirichlet}$, we could not reproduce the same performance as given in Malinin & Gales (2018) and hence use their reported results for CIFAR-10 for our comparison.

For MCDP, we use the standard DNN model with randomly dropping the nodes during test time. The predictive categorical distributions are obtained by averaging the outputs for 10 iterations.

ODIN applies the standard DNN models trained only using in-domain training examples for OOD detection. During testing phase, it perturbs the input images using FGSM adversarial attack (Goodfellow et al. (2014b))and softmax activation function by incorporating the temperature hyper-parameter (Hinton et al. (2015b)). The maximum Probability score is then applied for their uncertainty measure. They propose to use different hyper-parameters for different OOD examples. However, in practice, the source of expected OOD

examples cannot be known. Hence, for our comparisons, we always set the perturbation size to 0.002 and the temperature to 1000.

OE models are trained using the proposed loss function by Hendrycks et al. (2019). Here, we use the same training set up as applied for our $\text{DPN}_{soft}$ models: CIFAR-10 classifiers are trained using CIFAR-10 training images as in-domain examples and CIFAR-100 training images as OOD examples. For CIFAR-100, the OE models are trained using CIFAR-10 training images as OOD examples.

## D  DIFFERENTIAL ENTROPY MEASURE FOR DIRICHLET PRIOR NETWORK

Differential Entropy of a Dirichlet distribution can be calculated as follows (Malinin & Gales, 2018):

$$
\begin{aligned}
\mathcal{H}[p(\boldsymbol{\mu}|\boldsymbol{x}^*, D_{in})] &= -\int_{S^{K-1}} p(\boldsymbol{\mu}|\boldsymbol{x}^*, D_{in}) \ln p(\boldsymbol{\mu}|\boldsymbol{x}^*, D_{in}) \\
&= \sum_{c=1}^{K} \ln \Gamma(\alpha_c) - \ln \Gamma(\alpha_0) - \sum_{c=1}^{K} (\alpha_c - 1)(\psi(\alpha_c) - \psi(\alpha_0))
\end{aligned}
\tag{14}
$$

Note that, $\alpha_c$ is a function of $x^*$. $\Gamma$ and $\psi$ denotes the Gamma and digamma functions respectively.

