# OpenReview forum: "Improving Dirichlet Prior Network for Out-of-Distribution Example Detection"
_ICLR.cc/2020/Conference — Reject_

### Official Review · AnonReviewer3 · 2019-10-23
**Official Blind Review #3**

**Rating:** 3

**Review:**

This work studies the predictive uncertainty issue of deep learning models. In particular, this work focuses on the distributional uncertainty which is caused by distributional mismatch between training and test examples. The proposed method is developed based on the existing work called Dirichlet Prior Network (DPN). It aims to address the issue of DPN that its loss function is complicated and makes the optimization difficult. Instead, this paper proposes a new loss function for DPN, which consists of the commonly used cross-entropy loss term and a regularization term. Two loss functions are respectively defined over in-domain training examples and out-of-distribution (OOD) training examples. The final objective function is a weighted combination of the two loss functions. Experimental study is conducted on one synthetic dataset and two image datasets (CIFAR-10 and CIFAR-100) to demonstrate the properties of the proposed method and compare its performance with the relevant ones in the literature. The issue researched in this work is of significance because understanding the predictive uncertainty of a deep learning model has its both theoretical and practical value. The motivation, research issues and the proposed method are overall clearly presented.

The current recommendation is Weak Reject because the experimental study is not convincing or comprehensive enough.

1.	Although the goal of this work is to deal with the inefficiency issue of the objective function of existing DPN with the newly proposed one, this experimental study does not seem to conduct sufficient experiments to demonstrate the advantages (say, in terms of training efficiency & the capability in making the network scalable for more challenging dataset) of the proposed objective function over the existing one;
2.	Table 1 compares the proposed method with ODIN. However, as indicated in this work, ODIN is trained with in-domain examples only. Is this comparison fair? Actually, ODIN's setting seems to be more practical and more challenging than the setting used by the propose methods.
3.	The evaluation criteria shall be better explained at the beginning of the experiment, especially how they can be collectively used to verify that the proposed method can better distinguish distributional uncertainty from other uncertainty types.
4.	In addition, the experimental study can be clearer on the training and test splits. How many samples from CIFAR-10 and CIFAR-100 are used for training and test purpose, respectively? Also, since training examples are from CIFAR-10 and CIFAR-100 and the test examples are also from these two datasets, does this contradict with the motivation of “distributional mismatch between training and test examples” mentioned in the abstract?
5.	The experimental study can have more comparison on challenging datasets with more classes since it is indicated that DPN has difficulty in dealing with a large number of classes.

Minor:

1. Please define the \hat\theta in Eq.(5). Also, is the dirac delta estimation a good enough approximation here?
2. The \lambda_{out} < \lambda_{in} in Eq.(11) needs to be better explained. In particular, are the first terms in Eq.(10) and Eq.(11) comparable in terms of magnitude? Otherwise,  \lambda_{out} < \lambda_{in} may not make sense.
3. The novelty and significance of fine-tuning the proposed model with noisy OOD training images can be better justified.

**Experience Assessment:**

I have read many papers in this area.

**Review Assessment: Checking Correctness Of Derivations And Theory:**

I assessed the sensibility of the derivations and theory.

**Review Assessment: Checking Correctness Of Experiments:**

I carefully checked the experiments.

**Review Assessment: Thoroughness In Paper Reading:**

I read the paper at least twice and used my best judgement in assessing the paper.

---

> ### Author Response · Authors · 2019-11-09
> **Addressing Minor comments**
>
> 1. \hat \theta is a Dirac delta estimation of \theta (added in the revised draft). Since model uncertainty is reducible and can be explained away given enough data, we can use the Dirac delta approximation.
>
> 2. In our revised draft, we have added additional discussion to explain why we should choose \lambda_{out} < \lambda_{in}. Please refer to section 3.2 (see Fig. 2). This discussion was initially given in the experiment section. The first terms in eq. 10 and 11 are comparable. Both of them are representing the cross-entropy function.
>
> 3. “novelty and significance of fine-tuning the proposed model with noisy OOD training images”
>
> We have updated this discussion in sec 4.2: Data Augmentation with white-noise for Training. Hopefully, this is now properly understandable.

---

> ### Author Response · Authors · 2019-11-09
> **Response to Review #3**
>
> Thank you for your careful analysis and detailed feedback for our paper. We have addressed all your suggestions/concerns in our latest revised draft. We hope that our following response will clarify your concerns.
>
> Q1. “Experimental study does not seem to conduct sufficient experiments to demonstrate the advantages”
>
> The DPN framework [1] was able to train their model only for CIFAR-10, MNIST and SVHN dataset that contains only 10 image classes. In this paper, we have already shown that our proposed model can be scalable for CIFAR-100 images with 100 classes.
>
> We have now added results for TinyImageNet in Appendix A (Table 6 and 7). This dataset is a more challenging dataset with 200 classes. Please refer to Appendix C for the data description and model details.
>
> [1] Predictive uncertainty estimation via prior networks. NIPS. 2018.
>
> Q2. “ODIN is trained with in-domain examples only. Is this comparison fair?”
>
> Since the key objective of ODIN and our proposed DPN framework is the same i.e. to detect the OOD examples, we have provided a comparison of our models with ODIN. If the reviewer thinks that this is not a fair comparison, we can remove this comparison in our next revision.
>
> However, we do not agree with the comment that “ODIN's setting seems to be more practical and more challenging”. During the testing phase, the ODIN framework requires access to a smaller set of OOD test datasets for fine-tuning their models. This may not be possible in practice for real-world scenarios.
> In contrast, our proposed framework does not require any access to the OOD test-set.
>
> Q3. “how they can be collectively used to verify that the proposed method can better distinguish distributional uncertainty from other uncertainty types”
>
> We have addressed this comment in our updated draft. Please refer to section 4.2 (Detecting the source of uncertainty).
> Please note that the technique to “distinguishing distributional uncertainty from other uncertainty types” are different as we choose a positive or negative value for \lambda_out in our loss function during training.
>
> We have explained how we can determine the distributional uncertainty for a given test image by computing its entropy and D.Ent values.
>
> P.S.: We have moved the experiment with the Gaussian noise dataset in the appendix and included the misclassification detection task in the main paper for a better explanation of how our model can detect distributional uncertainty in a test input.
>
> Q4. “experimental study can be clearer” “CIFAR-10 and CIFAR-100 and the test examples are also from these two datasets”
>
> Please note that our models are trained only using CIFAR-10 and CIFAR-100 training images and do not use them during the testing phase. In our revised draft, we have added more emphasis that the OOD training data and the test sets are disjoint. Please refer to section 4.2, Table 1.
>
> Let us describe how we have trained our DPN models for the CIFAR-10 dataset.
>
> To train our CIFAR-10 models, we use the CIFAR-10 training set containing 50,000 images as our in-domain dataset. For OOD training images, we use CIFAR-100 training images containing 50,000 images.
> For fine-tuning these models with noisy OOD images, we again use CIFAR-100 training images and add random noise to them.
>
> During the testing phase, we use OOD datasets such as Tiny Imagenet, LSUN, etc and attempt to distinguish them from the in-domain CIFAR-10 test set (containing 10,000). These datasets remain unknown to our classifiers and we do not need any extra tuning for our model at the testing phase. The experimental setup of our framework is the same as [1,2].
>
> [1] Predictive uncertainty estimation via prior networks. NIPS. 2018.
> [2] Deep anomaly detection with outlier exposure. ICLR, 2019.
>
> Q4. Part2: "contradict with the motivation of distributional mismatch"
>
> It seems there may be a misunderstanding. Let us explain by considering the CIFAR-10 model again:
>
> During testing, we use the CIFAR-10 test images as an in-domain dataset and attempt to distinguish them from unknown OOD datasets (e.g. TinyImagenet, LSUN, Texture, etc.) and evaluate their distributional mismatch.
> Note that, these test images are disjoint from the training images applied to learn the model.
> Also, we do not use any CIFAR-100 images during testing the CIFAR-10 model. Please refer to Table 1 (or Table 8 in Appendix C).
>
> Q5. “The experimental study can have more comparison on challenging datasets with more classes”
>
> As we have mentioned earlier, the original DPN framework cannot be scaled for the CIFAR-100 dataset with 100 image classes. We have already shown that the DPN framework with our proposed loss function can be trained for CIFAR-100.
> We have now added results for TinyImageNet in Appendix A (Table 6 and 7).

---

### Official Review · AnonReviewer1 · 2019-10-24
**Official Blind Review #1**

**Rating:** 3

**Review:**

This paper proposes an improved DPN framework with a novel loss function, which uses the standard cross-entropy loss along with a regularization term to control the sharpness of the output Dirichlet distributions from the network.The proposed loss function aims to improve the training efficiency of the DPN framework for challenging classification tasks with large number of classes

The proposed improved DPN is very incremental. It only adds a simple regularization term to the standard cross-entropy loss. The regularization term is the precision of the Dirichlet from the DPN. The technical novelty and contribution is not significant.

The paper claims the proposed loss function allows distributional uncertainty to be modelled separately from data uncertainty and model uncertainty, and the proposed framework can improve efficiency. However these claims lack of sufficient support.

Moreover, determining the source of uncertainty is just a mean of achieving better classification model, not the goal. The experimental results are not very convincing in improving classification over OOD examples due to the lack of comparison with state-of-the-art related works. Note many domain adaptation methods can handle OOD examples.

**Experience Assessment:**

I do not know much about this area.

**Review Assessment: Checking Correctness Of Derivations And Theory:**

I did not assess the derivations or theory.

**Review Assessment: Checking Correctness Of Experiments:**

I assessed the sensibility of the experiments.

**Review Assessment: Thoroughness In Paper Reading:**

I made a quick assessment of this paper.

---

> ### Author Response · Authors · 2019-11-09
> **Response to Review #1**
>
> Thank you for your review and your questions. Please find our response in the following. Please let us know if you have any further concern or question.
>
> “Only adds a simple regularization term to the standard cross-entropy loss.”
>
> We would like to thank the reviewer for pointing out the simplicity of our proposed framework. While our proposed regularization term is simple, it is theoretically quite significant. It allows the outputs of a DNN classifier to be viewed as Dirichlet distribution where we can control the sharpness of the distribution for different types of inputs.
>
> Please note that the original DPN framework was proposed using a very complex loss function by incorporating KL-divergence between Dirichlet distributions. Due to such complex loss functions, DPN frameworks were difficult to scale for a classification task with more than 10 classes.
> By introducing our simple regularization term with the cross-entropy loss and reducing the complexity of the loss function, we show that our DPN frameworks can be scaled for 100’s of classes.
>
> “Claims lack of sufficient support: distributional uncertainty to be modeled separately from data uncertainty and model uncertainty”
>
> Please refer to Equation 3 in section 3.1 where we describe the DPN framework and how we can model distributional uncertainty separately from data uncertainty and model uncertainty. Our proposed loss function for a DPN framework is then gradually derived from this equation in section 3.2.
>
> Our synthetic-data experiment visually demonstrates that our proposed DPN framework can distinguish distributional uncertainty from other uncertainty types (See Figure 3). In our revised draft, we have described  how can we determine the source of uncertainty for an unknown test instance.
>
> “Experimental results are not very convincing in improving classification over OOD examples”
>
> The goal of this work is to detect the uncertainty and the source of uncertainty in a DNN classifier when it is misclassifying or has recieved an OOD example from an unknown distribution. Note that, these OOD images can belong to any distribution unknown to the classifier, not a specific OOD set.
> This is important to estimate uncertainty in sensitive real-world applications as it allows users to manually intervene and act in an informative way. This is an active research area with several works in recent years [1-8]. For our experiments, we have already compared our proposed frameworks with many of these recent works that aim to detect OOD examples for a DNN classifier.
>
> Also, we believe that improving the classification accuracy for OOD examples using domain adaptation techniques is a different task where the objective is to focus only on a specific set of OOD examples. This is not the objective of our work.
>
> [1] Dropout as a bayesian approximation: Representing model uncertainty in deep learning. ICML 2016
> [2] Simple and scalable predictive uncertainty estimation using deep ensembles. NIPS 2017
> [3] A baseline for detecting misclassified and out-of-distribution examples in neural networks, ICLR 2017
> [4] Predictive uncertainty estimation via prior networks, NIPS 2018
> [5] Training confidence-calibrated classifiers for detecting out-of-distribution samples, ICLR 2018
> [6] Enhancing the reliability of out-of-distribution image detection in neural networks. ICLR 2018
> [7] Deep anomaly detection with outlier exposure, ICLR 2019
> [8] Why relu networks yield highconfidence predictions far away from the training data and how to mitigate the problem, CVPR 2019

---

### Official Review · AnonReviewer4 · 2019-11-01
**Official Blind Review #4**

**Rating:** 6

**Review:**

The paper proposes a novel loss function using the standard cross-entropy loss along with a regularization term on logits for training the Dirichlet prior network. The benefit of using Dirichlet prior network is that it can distinguish the in-domain noisy data and completely out-of-domain data. For in-domain noisy data, the Dirichlet distribution should be sharp but in the middle of the simplex, while for the OOD data, the Dirichlet distribution should be flat. The Dirichlet prior network is proposed by Malinin & Gales (2018), the new method in this paper overcomes the challenge of training the network based on the KL divergence which cannot work well for dataset with large number of classes. The paper is well written and easy to follow. Here are some comments and questions:

- In the proposed loss function, could you explain what is the reason that you choose to use sigmoid(z_c(x)) instead of \sum exp(z_c(x))? As you mentioned in the paper,  \sum exp(z_c(x)) suggests the sharpness of the distribution. Shouldn’t using \sum exp(z_c(x)) be more direct than the sigmoid(z_c(x))?

- The methods requires OOD dataset for training. The authors used the same OOD dataset for training and test. One concern is that what if the OOD dataset for test is not available at training. What is the alternative plan? How does that perform?

- In Table 1, for Gaussian in-domain dataset, why is \sum exp(z_c(x)) able to distinguish Gaussian in-domain from original in-domain images? If I understand correctly, Gusaain in-domain should have a sharp distribution which means the differential entropy (D. Ent) is small (as illustrated in Figure 2(d)), and the sum of the exponential of logits (\sum exp(z_c(x))) should be large. For the original in-domain, it should also have a sharp distribution with small differential entropy and large sum of the exponential of logits. Then I don’t understand why in the table, the AUROC and AUPR for the measure \sum exp(z_c(x) are very high values while that for the measure D. Ent are very low.

- Could you also compute the sum of the exponential of logits for the synthetic data, since it is the only metric that is evaluated in the real data experiments but not in the synthetic data?

- Could you clarify how you compute the differential entropy of the Dirichlet distribution given an input x? Did you use \alpha_c = exp(z_c(x))?


**Experience Assessment:**

I have published one or two papers in this area.

**Review Assessment: Checking Correctness Of Derivations And Theory:**

I carefully checked the derivations and theory.

**Review Assessment: Checking Correctness Of Experiments:**

I carefully checked the experiments.

**Review Assessment: Thoroughness In Paper Reading:**

I read the paper at least twice and used my best judgement in assessing the paper.

---

> ### Author Response · Authors · 2019-11-09
> **Response to Review #4**
>
> Thank you for your careful analysis and detailed feedback for our paper. We have addressed most of your comments in our revised version. Please find our response in the following. Please let us know if you have any further concern or question.
>
> 1.    What is the reason that you choose to use sigmoid(z_c(x)) instead of \sum exp(z_c(x))?
>
> Sigmoid is a bounded function that always produces a value within the range of (0,1).
> On the other hand, \sum exp(z_c(x)) is unbounded function and can produce any value within the range of [0, \infty).
> Hence, if we choose \sum exp(z_c(x)) as our regularizer term in the loss function, it would become very difficult to optimize the network. (We have now modified Sec 3.2 (paragraph 2) to explain the choice of sigmoid function.)
>
> 2.    Same OOD dataset for training and test.
>
> We believe there may be a misunderstanding in the experimental setup. We have not used the same OOD dataset for training and test. In our revised draft, we have added more emphasis that the OOD training data and the test sets are disjoint (see Table 1).
>
> Let us describe how we have trained our DPN models for CIFAR-10 dataset:
> To train our CIFAR-10 models, we use CIFAR-10 training set containing 50,000 images as our in-domain dataset. For OOD training images, we use CIFAR-100 training images containing 50,000 images.
> For fine-tuning these models with noisy OOD images, we again use CIFAR-100 training images and add random noise to them.
>
> During the testing phase, we use OOD datasets such as Tiny Imagenet, LSUN, etc., and attempt to distinguish them from the in-domain CIFAR-10 test set (containing 10,000). These datasets remain unknown to our classifiers and we do not need any extra tuning for our model at the testing phase.
> The experimental setup of our framework is the same as [1,2].
>
> [1] Malinin, Andrey, and Mark Gales. "Predictive uncertainty estimation via prior networks." NIPS. 2018.
> [2] Hendrycks, Dan, Mantas Mazeika, and Thomas G. Dietterich. "Deep anomaly detection with outlier exposure." ICLR, 2019.
>
> 3.    The AUROC and AUPR for the measure \sum exp(z_c(x) are very high values while that for the measure D. Ent are very low.
>
> Please refer to the formula of D.Ent in Appendix-D. Note that, the computation of D.Ent does not only depend on the \sum exp(z_c(x)) but also depends on the individual values of exp (z_c (x))’s. Hence, it may be possible that even for the same value of \sum exp(z_c(x)), we get different for D.Ent values for different x’s.
> On the other hand, as we choose \lambda_out <0 for our training loss, our DPN network produces negative logit values for OOD examples. This leads to producing sharp Dirichlet distribution across the edge of the simplex and hence a high D.Ent score and low \sum exp(z_c(x)) value.
> Now, as we enforce the network to produce negative logit values for OOD examples, the network examples might also produce similar logit scores and eventually misclassified by the network.
>
> Hence, we may obtain a high AUROC and AUPR score for \sum exp(z_c(x) measure and a lower score for D. Ent.
>
> P.S.: We moved the experiments on Gaussian noise into the appendix and include the misclassification detection experiment in the main paper. Also, please note that we can consider the Gaussian-noise in-domain images as “in-domain” examples only when the models are not fine-tuned using noisy OOD training images.
>
> 4.    Compute the sum of the exponential of logits for the synthetic data:
> We have updated Figure 3 and added additional images to address this comment.
>
> 5.    How you compute the differential entropy? Did you use \alpha_c = exp(z_c(x))?
> The expression to compute the differential entropy is provided in Appendix D.
> Yes, it contains \alpha_c = exp(z_c(x)).

---

### Author Response · Authors · 2019-11-14
**Summary of the revised draft (Added results for TinyImageNet)**

We want to express our deep gratitude to all reviewers for their constructive suggestions for our paper. We have addressed all the concerns and suggestions in our revised draft and  reply to each reviewer in separate comments. The list of our changes are as follows:

1.    "OOD test datasets are different from the training datasets": (Reviewer  #4 and #3)
In our experiment (section 4.2), we have added more emphasis that the OOD training data and the test sets are disjoint. We include Table 1 to describe the training and test datasets for our experiments.

2.    "Distinguishing distinguish distributional uncertainty": (Reviewer #3)
In section 4.2, we have explained how our proposed framework can be used to determine the distributional uncertainty for a given test image by computing the entropy and D-Ent values.
We have included the misclassification detection task in the main paper to present a better explanation (See Table 2).

3.  "Results on more challenging datasets": (Reviewer #3)
In Appendix-A (Table 6 and 7), we have added results for the TinyImageNet dataset. TinyImageNet is a more challenging dataset containing natural images from 200 classes of size 64x64. Our DPN framework is also found to be scalable and effective for this dataset.

Training details for our TinyImageNet experiments can be found in Appendix-C.

---

### Decision · Program_Chairs · 2019-12-19

**Decision:**

Reject

**Comment:**

In this work the authors build on the Dirichlet prior network of Malinin & Gales, replacing the loss function and adding a regularization term which improve training in the setting with a significant number of classes.   Improving uncertainty for deep learning is a challenging but very important problem.  The reviewers of this paper gave two weak rejects (one is of low confidence) and one weak accept.  They found the paper well written, easy to follow and well motivated but somewhat incremental and not entirely empirically justified.  None of the reviewers were willing to strongly champion the paper for acceptance.  Unfortunately as such the paper falls below the bar for acceptance.  It appears that the authors significantly added to the experiments in the discussion phase and hopefully that will make the paper much stronger for a future submission.